# DOPAMINE: A RESEARCH FRAMEWORK FOR DEEP REINFORCEMENT LEARNING

## ABSTRACT

Deep reinforcement learning (deep RL) research has grown significantly in recent years. A number of software offerings now exist that provide stable, comprehensive implementations for benchmarking. At the same time, recent deep RL research has become more diverse in its goals. In this paper we introduce Dopamine, a new research framework for deep RL that aims to support some of that diversity. Dopamine is open-source, TensorFlow-based, and provides compact yet reliable implementations of some state-of-the-art deep RL agents. We complement this offering with a taxonomy of the different research objectives in deep RL research. While by no means exhaustive, our analysis highlights the heterogeneity of research in the field, and the value of frameworks such as ours.

## 1 INTRODUCTION

The field of deep reinforcement learning (RL), while relatively young, has already demonstrated its value, from decision-making from perception (Mnih et al., 2015) to superhuman game-playing (Silver et al., 2016) to robotic manipulation (Levine et al., 2016). While most of early RL research focused on illustrating specific aspects of decision-making, a defining characteristic of deep reinforcement learning research has been its emphasis on complex agent-environment interactions: for example, an agent navigating its way out of a video game maze, from vision, is a task difficult even for humans.

As a consequence of this shift towards more complex interactions, writing reusable software for deep RL research has also become more challenging. First, an "agent" is now a whole architecture: for example, OpenAI Baselines's implementation of Mnih et al.'s Deep Q-Network (DQN) agent is composed of 6 different modules (Dhariwal et al., 2017). Second, there is now a wealth of algorithms to choose from, so being comprehensive in one's implementation typically requires sacrificing simplicity. Most importantly, the growing diversity in deep RL research makes it difficult to foresee what software needs the next research project might have.

This paper introduces Dopamine, a new TensorFlow-based framework that aims to support fundamental deep RL research. Dopamine emphasizes being compact rather than comprehensive: the first version is made of 12 Python files. These provide tested implementations of state-of-the-art, value-based agents for the Arcade Learning Environment (Bellemare et al., 2013). The code is designed to be easily understood by newcomers to the field, yet performant enough for research at scale. To further facilitate research, we provide interactive notebooks, trained models, and downloadable training data for all of our agents – including reproductions of previously published learning curves.

Our design choices are guided by the idea that different research objectives have different software needs. We support this point by reviewing some of the major developments in value-based deep RL, in particular those derived from the DQN agent. We identify different research objectives and discuss expectations of code that supports these objectives: architecture research, comprehensive studies, visualization, algorithmic research, and instruction. We have engineered Dopamine with the last two of these objectives in mind and as such, we believe that it has a unique role to play in the deep RL frameworks ecosystem.

The paper is organized as follows. In Section 2 we identify common research objectives in deep reinforcement learning and discuss their particular software needs. In Section 3 we introduce Dopamine and argue for its effectiveness in addressing the needs surfaced in Section 2. In Section 4 we revisit

the guidelines put forth by Machado et al. (2018) and discuss Dopamine's adherence to them. In Section 5 we discuss related frameworks and provide concluding remarks in Section 6.

## 2 Software for Deep Reinforcement Learning Research

What makes a piece of code useful to research? The deep learning community has by now identified a number of operations critical to their research goals: component modularity, automatic differentiation, and visualization, to name a few (Abadi et al., 2016). Perhaps because it is a younger field, consensus on software has been elusive in deep RL. In this section we identify different aspects of deep RL research, supporting our categorization with an analysis of recent work. Our aim is not to be exhaustive, but to highlight the heterogeneous nature of this research. From this analysis, we argue that different research aims lead to different software considerations.

To narrow the scope of our question, we restrict our attention to a subset of deep RL research:

1. **Fundamental research** in deep reinforcement learning,
2. applied to or evaluated on **simulated environments**.

The software needs of commercial applications are likely to be significantly different from those of researchers; similarly, real-world environments typically require tailored software infrastructure. In this paper we focus on the Arcade Learning Environment (ALE) as a mature, well-understood environment for this kind of research; we believe this section's conclusions extend to similar environments, including continuous control (Duan et al., 2016; Tassa et al., 2018), first-person environments (Kempka et al., 2016; Beattie et al., 2016; Johnson et al., 2016), and to some extent real-time strategy games (Tian et al., 2017; Vinyals et al., 2017).

### 2.1 Case Study: The Deep Q-Network Architecture

We begin by taking a close look at the research genealogy of the deep Q-network agent (DQN). Through our review we identify distinct research objectives that recur in the field. DQN is a natural choice for this purpose: not only is it recognized as a milestone in deep reinforcement learning, but it has since been extensively studied and improved upon, providing us with a diversity of research work to study. We emphasize that the objectives we identify here are not mutually exclusive, and that our survey is by no means exhaustive – we highlight specific results because they are particularly clear examples.

What we call **architecture research** is concerned with the interaction between components, including network topologies, to create deep RL agents. DQN was innovative in its use of an agent architecture including target network, replay memory, and Atari-specific preprocessing. Since then, it has become commonplace, if not expected, that an agent is composed of multiple interacting components; consider A3C (Mnih et al., 2016), ACER (Wang et al., 2017), Reactor (Gruslys et al., 2018), and IMPALA (Espeholt et al., 2018).

**Algorithmic research** is concerned with improving the underlying algorithms that drive learning and behaviour in deep RL agents. Using the DQN architecture as a starting point, double DQN (van Hasselt et al., 2016) and gap-increasing methods (Bellemare et al., 2016a) both adapt the Q-Learning rule to be more statistically robust. Prioritized experience replay (Schaul et al., 2016) replaces the uniform sampling rule from the replay memory by one that more frequently samples states with high prediction error. Retrace($\lambda$) computes a sum of $n$-step returns, weighted by a truncated correction ratio derived from policy probabilities (Munos et al., 2016). The dueling algorithm (Wang et al., 2016) separates the estimation of Q-values into advantage and baseline components. Finally, distributional methods (Bellemare et al., 2017; Dabney et al., 2018b) replace the scalar prediction made by DQN with a value distribution, and accordingly introduce new losses to the algorithm. In our taxonomy, algorithmic research is not tied to a specific agent architecture.

**Comprehensive studies** look back at existing research to benchmark or otherwise study how existing methods perform under different conditions or hyperparameter settings. Hessel et al. (2018), as one well-executed example, provide an ablation study of the effect of six algorithmic improvements, among which are double DQN, prioritized experience replay, and distributional learning. This study, performed over 57 games from the ALE, provides definitive conclusions on the relative usefulness of

these improvements. Comprehensive studies compare the performance of many agent architectures, or of different established algorithms within a given architecture.

**Visualization** is concerned with gaining a deeper understanding of deep RL methods through focused interventions, typically with an emphasis away from these methods' performance. Guo et al. (2014) studied the visual patterns which maximally influenced hidden units within a DQN-like network. Mnih et al. (2015) performed a simple clustering analysis to understand the state dynamics within DQN. Zahavy et al. (2016) also used clustering methods within a simple graphical interface to study emerging state abstractions in DQN networks, and further visualized the saliency of various stimuli. Bellemare et al. (2016b) depicted the exploratory behaviour of intrinsically-motivated agents using maps from the game MONTEZUMA'S REVENGE. We note that this kind of research is not restricted to visual analysis *per se*.

## 2.2 DIFFERENT SOFTWARE FOR DIFFERENT OBJECTIVES

For each research objective identified above, we now ask: how are these objectives enabled by software? Specifically, we consider 1) the likelihood that the research can be completed using existing code, versus requiring researchers to write new code, 2) the shelf life of code produced during the course of the research, and 3) the value of high-performance code to the research. We shape our analysis along the axis of *code complexity*: how likely is it that the research objective require a complex (large, modular, abstract) framework? Conversely, how beneficial is it to pursue this kind of research in a simple (compact, monolithic, readily understood) framework?

**Comprehensive studies** are naturally implemented as variations within a common agent architecture (each of (Hessel et al., 2018)'s Rainbow agent's features can be enabled or disabled). Frameworks supporting a wide range of variations are *very likely to be complex*, because they unify diverse methods under common interfaces. On the other hand, many studies require no code beyond what the framework already provides, and any new code is likely to have a long shelf life as it plays a role in further studies or in new agents. Performance is critical because these studies are done at scale.

**Architecture research.** In software, what we call architecture research touches entire branches of code. Frameworks that support architecture research typically provide reusable modules (e.g., a general-purpose replay memory) and common interfaces, and as such are *likely to be complex*. Code from early iterations may be discarded, but eventually significant time and effort is spent to produce a stable product. This is especially likely when the research focuses on engineering issues, for example scaling up distributed training (Horgan et al., 2018).

As the examples above highlight, **visualization** is typically intrusive, but benefits from stable implementations of common tools (e.g., Tensorboard). As such, it may be best served by frameworks that *strike a balance between simplicity and complexity*. Keeping visualization code beyond the project for which it is developed is often onerous, and when visualization modules are reused they usually require additional engineering. Performance is rarely a concern.

**Algorithmic research** is realized in software at different scales, from a simple change in equation (double DQN) to new data structures (the sum tree in prioritized experience replay). We believe algorithmic research *benefits from simple frameworks*, by virtue that simple code makes it easier to implement radically different ideas. Algorithmic research typically requires multiple iterations from an initial design, and in our experience this iterative process leads to a significant amount of code being discarded. Performance is less of an issue, as the objective is often to demonstrate the feasibility or value of a new approach, rather than deploy it at scale.

Finally, a piece of research software may have an **instructional purpose**. By this we mean code that is developed not only to produce scientific results, but also to explain the methodology to others. In deep RL, there are relatively few publications with this stated intent; interactive notebooks have played much of the teaching role. We view this objective as *benefitting the most from simplicity*: teaching research software needs to be stable, trusted, and clear, all of which are facilitated by a small codebase. Teaching-minded code often sacrifices some performance to increase understandability, and may provide additional resources (notebooks, benchmark data, visualizations). This code is usually intended to have a long shelf life.

**Conclusions.** From the above taxonomy we conclude that there is a natural trade-off between simplicity and complexity in a deep RL research framework. The resulting choices empower some

research objectives, possibly at the detriment of others. As we explain below, Dopamine positions itself on the side of simplicity, aiming to facilitate algorithmic research and instructional purposes.

# 3 DOPAMINE

In this section we provide specifics of Dopamine, our TensorFlow-based framework. Dopamine is built to satisfy the following design principles:

- **Self-contained and compact:** Self-contained means the core logic is not contained in external, non-standard, libraries. A compact framework is one that contains a small number of files and lines of code. Software often satisfies one of these requirements at the expense of the other. Satisfying both results in a low barrier-of-entry for users, as a compact framework with minimal reliance on external libraries means it is necessary to go through only a small number of files to comprehend the framework's internals.
- **Reliable and reproducible:** A reliable framework is one that is demonstrably correct; in software engineering this is typically achieved via tests. A reproducible framework is one that facilitates the regeneration of published statistics, and makes novel scientific contributions easily shareable with the community.

Being self-contained and compact helps achieve our goal of providing a simple framework, and in turn supporting algorithmic research and instructional purposes. By putting special emphasis on reliability, we also ensure that the resulting research can be trusted – acknowledging recent concerns on reproducibility in deep reinforcement learning (Islam et al., 2017; Henderson et al., 2017).

The initial offering of Dopamine focuses on value-based reinforcement learning applied to the Arcade Learning Environment. This restricted scope allowed us to make critical design decisions to achieve our goal of designing a simple framework. We intend for future expansions (see Section 6 for details) to follow the guidelines we set here. Although the focus of Dopamine is not computational performance, we provide some statistics in Appendix A.

## 3.1 SELF-CONTAINED AND COMPACT

Dopamine's design is illustrated in Figure 1, highlighting the main components and the lifecycle of an experiment. As indicated in the figure, our complete codebase consists of 12 files containing a little over 2000 lines of Python code.

The Runner class manages the interaction between the agent and the ALE (e.g., taking steps and receiving observations) as well as the bookkeeping (e.g., checkpointing and logging via the Checkpointer and Logger, respectively). The Checkpointer is in charge of regularly saving the experiment state which enables graceful recovery after failure, as well as learned weight re-use. The Logger is in charge of saving experiment statistics (e.g., accumulated training or evaluation rewards) to disk for visualization. We provide Colab interactive notebooks to facilitate the visualization of these statistics.

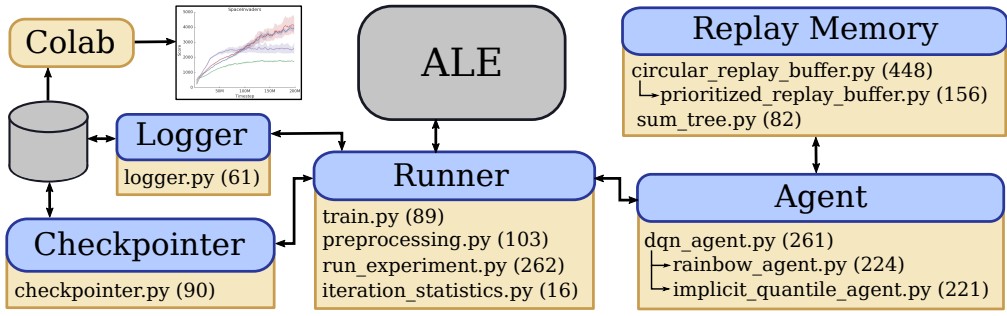

Figure 1: The design of Dopamine. Blue boxes are software components. Yellow boxes indicate the file names composing the software components; directed arrows indicate class inheritance, while the numbers in parentheses indicate the number of non-comment Python lines.

```
1 DQNAgent.gamma = 0.99
2 DQNAgent.epsilon_train = 0.01
3 DQNAgent.epsilon_decay_period = 250000   # agent steps
4 DQNAgent.optimizer = @tf.train.RMSPropOptimizer()
5 tf.train.RMSPropOptimizer.learning_rate = 0.00025
6 Runner.sticky_actions = True
7 WrappedReplayBuffer.replay_capacity = 1000000
8 WrappedReplayBuffer.batch_size = 32
```

Figure 2: A few lines from the DQN gin-config file implementing the default Dopamine settings.

The different agent classes contain the core logic for inference and learning: receiving Atari 2600 frames from the Runner, returning actions to perform, and performing learning. As in Mnih et al. (2015), the agents make use of a replay memory for the learning process. We provide complete implementations of four established agents: DQN (Mnih et al., 2015) (which is the base class for all agents), C51 (Bellemare et al., 2017), a simplified version of the single-GPU Rainbow agent (Hessel et al., 2018), and IQN (Dabney et al., 2018a). Our version of the Rainbow agent includes the three components identified as most important by Hessel et al. (2018): $n$-step updates (Mnih et al., 2016), prioritized experience replay (Schaul et al., 2016), and distributional reinforcement learning (Bellemare et al., 2017). In particular, our version of Rainbow does not include double DQN, the dueling architecture, or noisy networks (details in the original work). In our codebase, C51 is a particular parametrization of the Rainbow agent.

In order to highlight the simplicity with which one can create new agents, we provide some code examples in Appendices B and C. The code provided demonstrates how one can perform modifications to one of the provided agents or create a new one from scratch.

### 3.2 RELIABLE AND REPRODUCIBLE

We provide a complete suite of tests for all our codebase with code coverage of over 98%. In addition to helping ensure the correctness of the code, these tests provide an alternate form of documentation, complementing the regular documentation and interactive notebooks provided with the framework.

Dopamine makes use of gin-config (`github.com/google/gin-config`) for configuration of the different modules. Gin-config is a simple scheme for parameter injection, i.e. changing the default parameters of a method dynamically. In Dopamine we specify all parameters of an experiment within a single file. Figure 2 shows a sample of the configuration of the default DQN agent settings (full gin-config files for all agents are provided in Appendix D).

### 3.3 BASELINES FOR COMPARISON

We also provide a set of pre-trained baselines for the community to benchmark against. We use a uniform set of hyperparameters for all the provided agents, and call these the default settings. These combine Hessel et al. (2018)'s agent hyperparameters with Machado et al. (2018)'s ALE parameters. Our intent is not to provide an optimal set of hyperparameters, but to provide a *consistent* set as a baseline, while facilitating the process of hyperparameter exploration. Table 1 summarizes the

|  | **Dopamine** | **DQN** | **C51** | **Rainbow** | **IQN** |
|---|---|---|---|---|---|
| **Sticky actions** | Yes | No | No | No | No |
| **Epis. termination** | Game Over | Life Loss | Life Loss | Life Loss | Life Loss |
| **Training $\epsilon$** | 0.01 | 0.1 | 0.01 | 0.01 | 0.01 |
| **Evaluation $\epsilon$** | 0.001 | 0.01 | 0.001 | 0.001 | 0.001 |
| **$\epsilon$ decay schedule (frames)** | 1M | 4M | 4M | 1M | 4M |
| **Min. history to learn (frames)** | 80K | 200K | 200K | 80K | 200K |
| **Target net. update freq. (frames)** | 32K | 40K | 40K | 32K | 40K |

Table 1: Comparison of the hyperparameters used by published agents and our default settings.

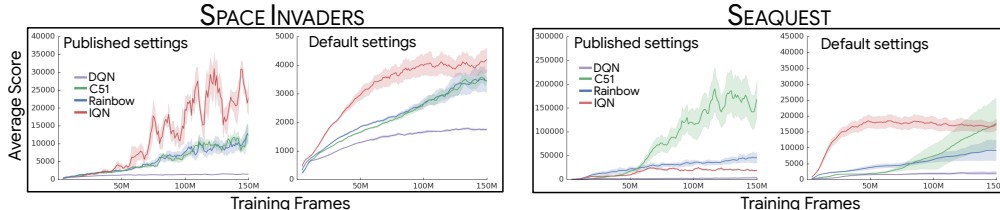

Figure 3: Comparing results of the published settings against the default settings, averaged over 5 runs. In each game, note that the y-scales between published and default settings are different; this is due mostly to the use of sticky actions in our default setting. The relative dynamics between the different algorithms appear unchanged for SpaceInvaders, but quite different in Seaquest.

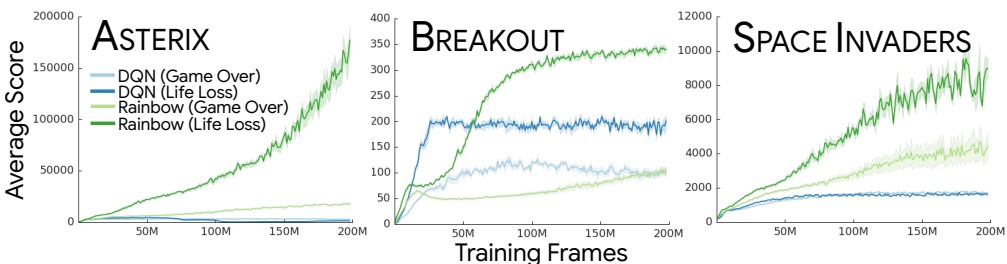

Figure 4: Effect of terminating episodes on Life Loss or Game Over, averaged over 5 runs; this choice can have a dramatic difference in reported performance, although less so for DQN.

differences between our default experimental setup and published results. For completeness, our experiments also use the minimal action set from ALE, as this is what the Gym interface provides.

For each agent, we include gin-config files for both the default and published settings. In Figure 3 we compare the four agents' published settings against the default settings on SPACE INVADERS and SEAQUEST. Note that the change in scale in the y-axis is due to the use of sticky actions. It is interesting to note the changing dynamics between the algorithms in Seaquest when going from their published settings to the default settings. With the former, C51 dominates the other algorithms by a large margin, and Rainbow dominates over IQN by a smaller margin. With the latter, IQN seems to dominate over all algorithms from early on.

To facilitate benchmarking against our default settings we ran 5 independent runs of each agent on all 60 games from the ALE. For each of these runs we provide the TensorFlow checkpoints for all agents, the event files for Tensorboard, the training logs for all of the agents, a set of interactive notebooks facilitating the plotting of new experiments against our baselines, as well as a webpage where one can visually inspect the four agents' performance across all 60 games.

## 4    REVISITING THE ARCADE LEARNING ENVIRONMENT: A TEST CASE

Machado et al. (2018) propose a standard methodology for evaluating algorithms within the ALE, and provide empirical evidence that alternate ALE parameter choices can impact research conclusions. In this section we continue the investigation where they left off: we begin with DQN and look ahead to some of the algorithms it inspired; namely, C51 (Bellemare et al., 2017) and Rainbow (Hessel et al., 2018). For legibility we are plotting DQN against C51 or Rainbow, but not both; the qualitative results, however, remain the same when compared with either. For continuity, we employ the parameter names from Section 3.1 from Machado et al.'s work.

### 4.1    EPISODE TERMINATION

The ALE considers an episode as finished when a human would normally stop playing: when they have finished the game or run out of lives. We call this termination condition "Game Over". Mnih et al. (2015) introduced a heuristic, called *Life Loss*, which adds artificial episode boundaries in the

replay memory whenever the player loses a life (e.g., in MONTEZUMA'S REVENGE the agent has 5 lives). Both definitions of episode termination have been used in the recent literature. Running this experiment in Dopamine consists in modifying the following gin-config option:

```
AtariPreprocessing.terminal_on_life_loss = True
```

Figure 4 illustrates the difference in reported performance under the two conditions. Although our plots show that the Life Loss heuristic improves performance in some of the simpler games, Bellemare et al. (2016b) pointed out that it hinders performance in others, in particular because the agent cannot learn about the true consequences of losing a life. Following Machado et al.'s guidelines, the Life Loss heuristic is disabled in the default settings.

## 4.2 MEASURING TRAINING DATA AND SUMMARIZING LEARNING PERFORMANCE

In Dopamine, training data is measured in game frames experienced by the agent; and each iteration consists in a fixed number of frames, rounded up to the nearest episode boundary.

Dopamine supports two schedules for running jobs: `train` and `train_and_eval`. The former only measures average score during training, while the latter interleaves these with evaluation runs, where learning is stopped. Machado et al. (2018) advocate for reporting the learning curves during training, necessitating only the `train` schedule. We report the difference in reported scores between the training and evaluation scores in Figure 5. This graph suggests that there is little difference between reporting training versus evaluation returns, so restricting to training curves is sufficient.

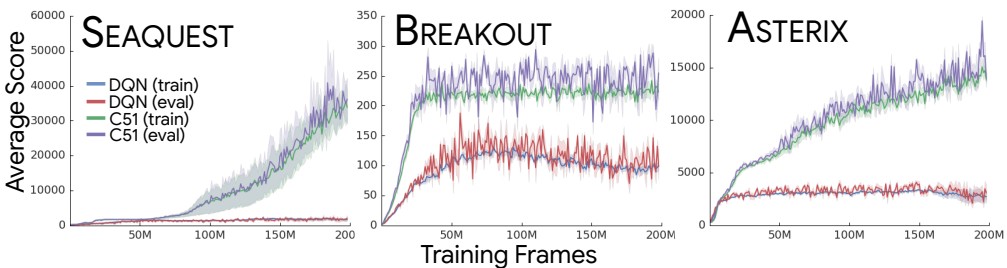

Figure 5: Training and evaluation scores, averaged over 5 runs. For our default values of training and evaluation $\epsilon$, there is a minimal difference between the two settings.

## 4.3 EFFECT OF STICKY ACTIONS ON AGENT PERFORMANCE

The original ALE has deterministic transitions, which rewards agents that can memorize sequences of actions to achieve high scores (e.g., The Brute in Machado et al., 2018). To mitigate this issue, the most recent version of the ALE implements *sticky actions*. Sticky actions make use of a *stickiness parameter* $\varsigma$, which is the probability that the environment will execute the agent's *previous* action, as opposed to the one the agent just selected – effectively implementing a form of action momentum. Running this experiment in Dopamine consisted in modifying the following gin-config option:

```
Runner.sticky_actions = False
```

In Figure 6 we demonstrate that there are differences in performance when running with or without sticky actions. While in some cases (Rainbow playing SPACE INVADERS) sticky actions seem to improve performance, they do typically reduce performance. Nevertheless, they still lead to meaningful learning curves (Rainbow surpassing DQN); hence, and in accordance with the recommendations given by Machado et al. (2018), sticky actions are enabled by default in Dopamine.

## 5 RELATED WORK

We acknowledge that the last two years have seen the multiplication of frameworks for deep RL, most targeting fundamental research. This section reviews some of the more popular of these, and

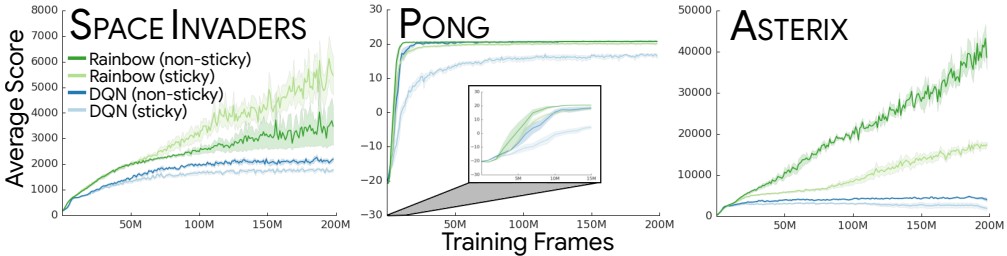

Figure 6: Effect of sticky vs. non-sticky actions. Sticky actions affect the performance of both Rainbow and DQN, but preserve the qualitative differences in performance between the two.

provides our perspective on their interplay with the taxonomy of Section 2. From this review, we conclude that Dopamine fills a unique niche in the deep reinforcement learning software ecosystem.

OpenAI baselines (Dhariwal et al., 2017) is a popular library providing comprehensive implementations of deep RL algorithms, with a particular focus on single-threaded, policy-based algorithms. Similar libraries include Coach from Intel (Caspi et al., 2017), Tensorforce (Schaarschmidt et al., 2017), and Keras RL (Plappert, 2016), which provide state-of-the-art algorithms and easy integration with OpenAI Gym. RLLab (Duan et al., 2016) emphasizes the benchmarking of continuous control algorithms, while RLLib (Liang et al., 2018) focuses on defining abstractions to optimize reinforcement learning performance in distributed settings. ELF (Tian et al., 2017) is a distributed RL framework in C++ and Python focusing on real-time strategy games. By virtue of providing implementations for many agents and algorithms, these frameworks are well-positioned to perform architecture research and comprehensive studies.

The field has also benefited from the open-sourcing of individual agents, typically done to facilitate reproducibility and disseminate technological *savoir-faire*. Of note, DQN was originally open-sourced in Lua and has since been re-implemented countless times; similarly, there are a number of publicly available PPO implementations (e.g. Hafner et al., 2017). More recently, the IMPALA agent was also open-sourced (Espeholt et al., 2018). While still beneficial, these frameworks are typically intended for personal consumption or illustrative purposes.

We conclude by noting that numerous reinforcement learning frameworks have been developed prior and in parallel to the deep RL resurgence, with similar research objectives. RLGlue (Tanner & White, 2009) emphasized benchmarking of value-based algorithms across a variety of canonical tasks, and drove a yearly competition; BURLAP (MacGlashan, 2016) provides object-oriented support; PyBrain (Schaul et al., 2010) supports neural networks, and is perhaps closest to more recent frameworks; RLPy (Geramifard et al., 2015) is a lighter framework written in Python, with similar goals to RLGlue and also emphasizing instructional purposes.

## 6    CONCLUSION AND FUTURE WORK

Dopamine provides a stable, reliable, flexible, and reproducible framework for fundamental deep reinforcement learning research. In this paper we have highlighted some of the challenges the research community faces in making new research reproducible and easy to compare against, and argue how Dopamine addresses many of these issues. Our hope is that by providing the community with a stable framework that is not difficult to understand will propel new scientific advances in this continually-growing field of research.

In order to keep our initial offering as simple and compact as possible, we have focused on single-GPU, value-based agents running on the ALE. In the near future, we hope to extend these offerings to policy-based methods as well as other environments. Distributed methods are an avenue that we are considering, although we are approaching it with caution in order to avoid introducing extra complexity into the codebase. Finally, we would like to provide tools for more advanced visualizations such as those discussed in Section 2.2.

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

## A   PERFORMANCE STATISTICS FOR DOPAMINE

Although the focus of Dopamine is not computational performance, we provide some performance statistics below.

### A.1   RUNTIME

Runtime performance is normally measured in the number of Atari frames processed per second (fps). This performance will vary from agent to agent and from game to game, but we provide two extremes of this spectrum. Both were measured while training on a Tesla P100 GPU.

- **DQN on Pong:** Runtime is around 800 fps.
- **IQN on Asterix:** Runtime is around 371 fps.

### A.2   DISKSPACE

As mentioned in the main body of the text, Dopamine performs regular checkpointing to allow for graceful recovery from failures. We do perform garbage collection, so maintain checkpoints only for the last few iterations. The largest consumer of diskspace is the replay buffer, as it is configured by default to hold 1 million frames. When checkpointing, we store these as compressed numpy objects. Once again, the footprint varies not only from agent to agent and from game to game, but also from one run to the next, as the complexity of the frames stored depend on the policy learned by the agent. We provide two extremes of this spectrum:

- **DQN on Pong:** The compressed replay buffer can be as low as 4.3Kb. Pong frames have few "active" pixels per frame, so this is not surprising. See screenshot below:

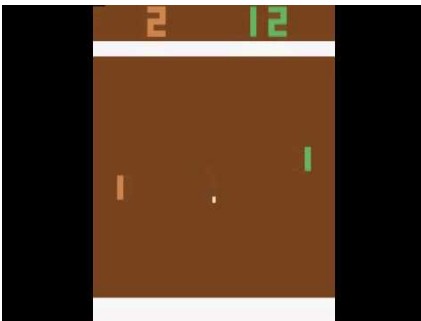

- **IQN on YarsRevenge:** The compressed replay buffer can be as high as 1.2Gb. YarsRevenge has a pseudo-random strip running down the screen, so this is also not surprising. See screenshot below:

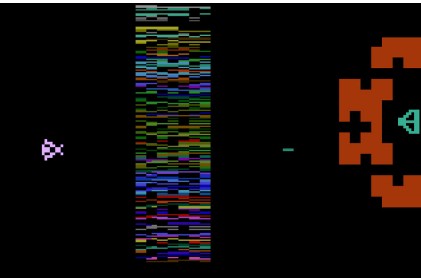

## B    MODIFYING DQN TO SELECT RANDOM ACTIONS

In this section we demonstrate how one can create a new agent by inheriting from one of the provided agents. This code is meant solely for illustrative purposes, as the created agent will perform quite poorly.

```python
import numpy as np
import os
from dopamine.agents.dqn import dqn_agent
from dopamine.atari import run_experiment
from dopamine.colab import utils as colab_utils
from absl import flags

BASE_PATH = '/tmp/colab_dope_run'
GAME = 'Asterix'

LOG_PATH = os.path.join(BASE_PATH, 'random_dqn', GAME)

class MyRandomDQNAgent(dqn_agent.DQNAgent):
  def __init__(self, sess, num_actions):
    """This maintains all the DQN default argument values."""
    super(MyRandomDQNAgent, self).__init__(sess, num_actions)

  def step(self, reward, observation):
    """Calls the step function of the parent class, but returns a random
    action.
    """
    _ = super(MyRandomDQNAgent, self).step(reward, observation)
    return np.random.randint(self.num_actions)

def create_random_dqn_agent(sess, environment):
  """The Runner class will expect a function of this type to create an
    agent."""
  return MyRandomDQNAgent(sess, num_actions=environment.action_space.n)

# Create the runner class with this agent. We use very small numbers of
    steps
# to terminate quickly, as this is mostly meant for demonstrating how one
    can
# use the framework. We also explicitly terminate after 110 iterations (
    instead
# of the standard 200) to demonstrate the plotting of partial runs.
random_dqn_runner = run_experiment.Runner(LOG_PATH,
                                          create_random_dqn_agent,
                                          game_name=GAME,
                                          num_iterations=200,
                                          training_steps=10,
                                          evaluation_steps=10,
                                          max_steps_per_episode=100)

print('Will train agent, please be patient, may be a while...')
random_dqn_runner.run_experiment()
print('Done training!')
```

This code snippet is also provided in one of our interactive notebooks, where users can train and visualize the agent's performance against the trained baselines as follows:

```python
import seaborn as sns
import matplotlib.pyplot as plt

random_dqn_data = colab_utils.read_experiment(LOG_PATH, verbose=True)
random_dqn_data['agent'] = 'MyRandomDQN'
random_dqn_data['run_number'] = 1
experimental_data[GAME] = experimental_data[GAME].merge(random_dqn_data,
```

```
 8                                                   how='outer')
 9
10 fig, ax = plt.subplots(figsize=(16,8))
11 sns.tsplot(data=experimental_data[GAME], time='iteration', unit='
       run_number',
12             condition='agent', value='train_episode_returns', ax=ax)
13 plt.title(GAME)
14 plt.show()
```

Resulting in the following plot:

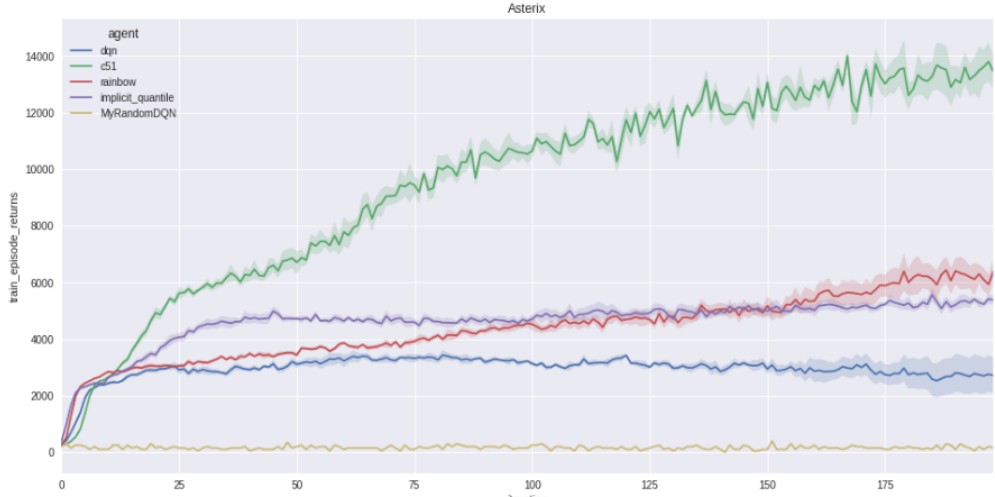

# C    CREATING A NEW AGENT FROM SCRATCH

In this section we demonstrate how to create a new agent from scratch by providing the minimum functionality expected by the Runner. Again, this code is meant solely for illustrative purposes.

```python
LOG_PATH = os.path.join(BASE_PATH, 'sticky_agent', GAME)

class StickyAgent(object):
  """This agent randomly selects an action and sticks to it. It will
     change
  actions with probability switch_prob."""
  def __init__(self, sess, num_actions, switch_prob=0.1):
    self._sess = sess
    self._num_actions = num_actions
    self._switch_prob = switch_prob
    self._last_action = np.random.randint(num_actions)
    self.eval_mode = False

  def _choose_action(self):
    if np.random.random() <= self._switch_prob:
      self._last_action = np.random.randint(self._num_actions)
    return self._last_action

  def bundle_and_checkpoint(self, unused_checkpoint_dir, unused_iteration
    ):
    pass

  def unbundle(self, unused_checkpoint_dir, unused_checkpoint_version,
               unused_data):
    pass

  def begin_episode(self, unused_observation):
    return self._choose_action()

  def end_episode(self, unused_reward):
    pass

  def step(self, reward, observation):
    return self._choose_action()

def create_sticky_agent(sess, environment):
  """The Runner class will expect a function of this type to create an
     agent."""
  return StickyAgent(sess, num_actions=environment.action_space.n,
                     switch_prob=0.2)

# Create the runner class with this agent. We use very small numbers of
     steps
# to terminate quickly, as this is mostly meant for demonstrating how one
     can
# use the framework. We also explicitly terminate after 110 iterations (
     instead
# of the standard 200) to demonstrate the plotting of partial runs.
sticky_runner = run_experiment.Runner(LOG_PATH,
                                      create_sticky_agent,
                                      game_name=GAME,
                                      num_iterations=200,
                                      training_steps=10,
                                      evaluation_steps=10,
                                      max_steps_per_episode=100)

print('Will train sticky agent, please be patient, may be a while...')
sticky_runner.run_experiment()
print('Done training!')
```

This code snippet is also provided in one of our interactive notebooks, where users can train and visualize the agent's performance against the trained baselines as follows:

```
import seaborn as sns
import matplotlib.pyplot as plt

sticky_data = colab_utils.read_experiment(log_path=LOG_PATH, verbose=True
    )
sticky_data['agent'] = 'StickyAgent'
sticky_data['run_number'] = 1
experimental_data[GAME] = experimental_data[GAME].merge(sticky_data,
                                                how='outer')

fig, ax = plt.subplots(figsize=(16,8))
sns.tsplot(data=experimental_data[GAME], time='iteration', unit='
    run_number',
            condition='agent', value='train_episode_returns', ax=ax)
plt.title(GAME)
plt.show()
```

Resulting in the following plot:

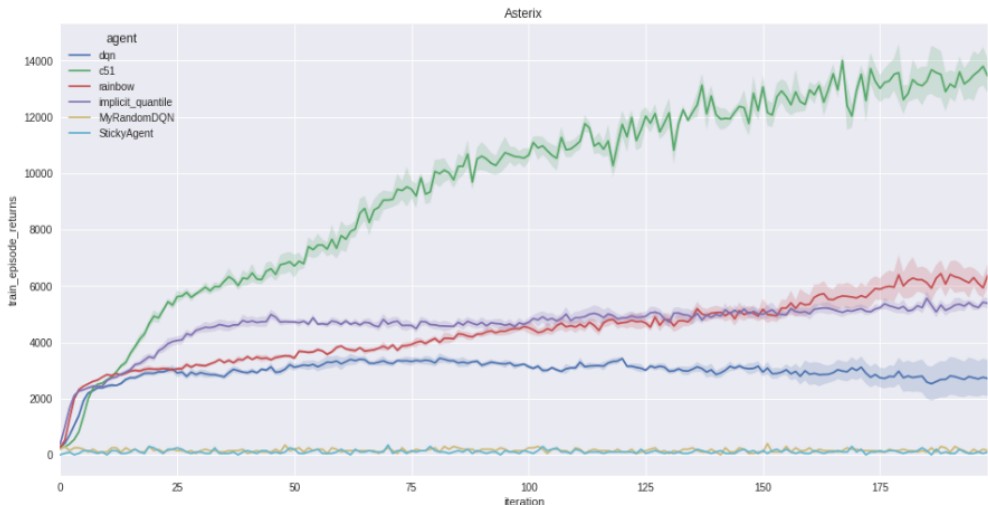

## D  GIN CONFIG FILES FOR ALL AGENTS

In this section we list all the gin config files provided for all our agents.

### D.1  DQN

#### D.1.1  DEFAULT SETTINGS

Default settings used in Dopamine:

```
DQNAgent.gamma = 0.99
DQNAgent.update_horizon = 1
DQNAgent.min_replay_history = 20000  # agent steps
DQNAgent.update_period = 4
DQNAgent.target_update_period = 8000  # agent steps
DQNAgent.epsilon_train = 0.01
DQNAgent.epsilon_eval = 0.001
DQNAgent.epsilon_decay_period = 250000  # agent steps
DQNAgent.tf_device = '/gpu:0'  # use '/cpu:*' for non-GPU version
DQNAgent.optimizer = @tf.train.RMSPropOptimizer()
```

```
11
12 tf.train.RMSPropOptimizer.learning_rate = 0.00025
13 tf.train.RMSPropOptimizer.decay = 0.95
14 tf.train.RMSPropOptimizer.momentum = 0.0
15 tf.train.RMSPropOptimizer.epsilon = 0.00001
16 tf.train.RMSPropOptimizer.centered = True
17
18 Runner.game_name = 'Pong'
19 Runner.sticky_actions = True
20 Runner.num_iterations = 200
21 Runner.training_steps = 250000  # agent steps
22 Runner.evaluation_steps = 125000  # agent steps
23 Runner.max_steps_per_episode = 27000  # agent steps
24
25 WrappedReplayBuffer.replay_capacity = 1000000
26 WrappedReplayBuffer.batch_size = 32
```

### D.1.2   NATURE SETTINGS

Settings used in Mnih et al. (2015):

```
1 # Hyperparameters used in Mnih et al. (2015).
2 import dopamine.atari.preprocessing
3 import dopamine.atari.run_experiment
4 import dopamine.agents.dqn.dqn_agent
5 import dopamine.replay_memory.circular_replay_buffer
6 import gin.tf.external_configurables
7
8 DQNAgent.gamma = 0.99
9 DQNAgent.update_horizon = 1
10 DQNAgent.min_replay_history = 50000  # agent steps
11 DQNAgent.update_period = 4
12 DQNAgent.target_update_period = 10000  # agent steps
13 DQNAgent.epsilon_train = 0.1
14 DQNAgent.epsilon_eval = 0.05
15 DQNAgent.epsilon_decay_period = 1000000  # agent steps
16 DQNAgent.tf_device = '/gpu:0'  # use '/cpu:*' for non-GPU version
17 DQNAgent.optimizer = @tf.train.RMSPropOptimizer()
18
19 tf.train.RMSPropOptimizer.learning_rate = 0.00025
20 tf.train.RMSPropOptimizer.decay = 0.95
21 tf.train.RMSPropOptimizer.momentum = 0.0
22 tf.train.RMSPropOptimizer.epsilon = 0.00001
23 tf.train.RMSPropOptimizer.centered = True
24
25 Runner.game_name = 'Pong'
26 # Deterministic ALE version used in the DQN Nature paper (Mnih et al.,
     2015).
27 Runner.sticky_actions = False
28 Runner.num_iterations = 200
29 Runner.training_steps = 250000  # agent steps
30 Runner.evaluation_steps = 125000  # agent steps
31 Runner.max_steps_per_episode = 27000  # agent steps
32
33 AtariPreprocessing.terminal_on_life_loss = True
34
35 WrappedReplayBuffer.replay_capacity = 1000000
36 WrappedReplayBuffer.batch_size = 32
```

### D.1.3   ICML SETTINGS

Settings used in Bellemare et al. (2017):

```
1  # Hyperparameters used for reporting DQN results in Bellemare et al.
       (2017).
2  import dopamine.atari.run_experiment
3  import dopamine.agents.dqn.dqn_agent
4  import dopamine.replay_memory.circular_replay_buffer
5  import gin.tf.external_configurables
6
7  DQNAgent.gamma = 0.99
8  DQNAgent.update_horizon = 1
9  DQNAgent.min_replay_history = 50000  # agent steps
10 DQNAgent.update_period = 4
11 DQNAgent.target_update_period = 10000  # agent steps
12 DQNAgent.epsilon_train = 0.01
13 DQNAgent.epsilon_eval = 0.001
14 DQNAgent.epsilon_decay_period = 1000000  # agent steps
15 DQNAgent.tf_device = '/gpu:0'  # use '/cpu:*' for non-GPU version
16 DQNAgent.optimizer = @tf.train.RMSPropOptimizer()
17
18 tf.train.RMSPropOptimizer.learning_rate = 0.00025
19 tf.train.RMSPropOptimizer.decay = 0.95
20 tf.train.RMSPropOptimizer.momentum = 0.0
21 tf.train.RMSPropOptimizer.epsilon = 0.00001
22 tf.train.RMSPropOptimizer.centered = True
23
24 Runner.game_name = 'Pong'
25 # Deterministic ALE version used in the DQN Nature paper (Mnih et al.,
       2015).
26 Runner.sticky_actions = False
27 Runner.num_iterations = 200
28 Runner.training_steps = 250000  # agent steps
29 Runner.evaluation_steps = 125000  # agent steps
30 Runner.max_steps_per_episode = 27000  # agent steps
31
32 WrappedReplayBuffer.replay_capacity = 1000000
33 WrappedReplayBuffer.batch_size = 32
```

## D.2  C51

### D.2.1  DEFAULT SETTINGS

Default settings used in Dopamine:

```
1  # Hyperparameters follow the settings from Bellemare et al. (2017), but
       we
2  # modify as necessary to match those used in Rainbow (Hessel et al.,
       2018), to
3  # ensure apples-to-apples comparison.
4  import dopamine.agents.rainbow.rainbow_agent
5  import dopamine.atari.run_experiment
6  import dopamine.replay_memory.prioritized_replay_buffer
7  import gin.tf.external_configurables
8
9  RainbowAgent.num_atoms = 51
10 RainbowAgent.vmax = 10.
11 RainbowAgent.gamma = 0.99
12 RainbowAgent.update_horizon = 1
13 RainbowAgent.min_replay_history = 20000  # agent steps
14 RainbowAgent.update_period = 4
15 RainbowAgent.target_update_period = 8000  # agent steps
16 RainbowAgent.epsilon_train = 0.01
17 RainbowAgent.epsilon_eval = 0.001
18 RainbowAgent.epsilon_decay_period = 250000  # agent steps
19 RainbowAgent.replay_scheme = 'uniform'
20 RainbowAgent.tf_device = '/gpu:0'  # use '/cpu:*' for non-GPU version
```

```
21 RainbowAgent.optimizer = @tf.train.AdamOptimizer()
22
23 tf.train.AdamOptimizer.learning_rate = 0.00025
24 tf.train.AdamOptimizer.epsilon = 0.0003125
25
26 Runner.game_name = 'Pong'
27 # Sticky actions with probability 0.25, as suggested by (Machado et al.,
       2017).
28 Runner.sticky_actions = True
29 Runner.num_iterations = 200
30 Runner.training_steps = 250000  # agent steps
31 Runner.evaluation_steps = 125000  # agent steps
32 Runner.max_steps_per_episode = 27000  # agent steps
33
34 WrappedPrioritizedReplayBuffer.replay_capacity = 1000000
35 WrappedPrioritizedReplayBuffer.batch_size = 32
```

### D.2.2  ICML SETTINGS

Settings used in Bellemare et al. (2017):

```
1 # Hyperparameters used in Bellemare et al. (2017).
2 import dopamine.atari.preprocessing
3 import dopamine.agents.rainbow.rainbow_agent
4 import dopamine.atari.run_experiment
5 import dopamine.replay_memory.prioritized_replay_buffer
6 import gin.tf.external_configurables
7
8 RainbowAgent.num_atoms = 51
9 RainbowAgent.vmax = 10.
10 RainbowAgent.gamma = 0.99
11 RainbowAgent.update_horizon = 1
12 RainbowAgent.min_replay_history = 50000  # agent steps
13 RainbowAgent.update_period = 4
14 RainbowAgent.target_update_period = 10000  # agent steps
15 RainbowAgent.epsilon_train = 0.01
16 RainbowAgent.epsilon_eval = 0.001
17 RainbowAgent.epsilon_decay_period = 1000000  # agent steps
18 RainbowAgent.replay_scheme = 'uniform'
19 RainbowAgent.tf_device = '/gpu:0'  # use '/cpu:*' for non-GPU version
20 RainbowAgent.optimizer = @tf.train.AdamOptimizer()
21
22 tf.train.AdamOptimizer.learning_rate = 0.00025
23 tf.train.AdamOptimizer.epsilon = 0.0003125
24
25 Runner.game_name = 'Pong'
26 # Deterministic ALE version used in the DQN Nature paper (Mnih et al.,
       2015).
27 Runner.sticky_actions = False
28 Runner.num_iterations = 200
29 Runner.training_steps = 250000  # agent steps
30 Runner.evaluation_steps = 125000  # agent steps
31 Runner.max_steps_per_episode = 27000  # agent steps
32
33 AtariPreprocessing.terminal_on_life_loss = True
34
35 WrappedPrioritizedReplayBuffer.replay_capacity = 1000000
36 WrappedPrioritizedReplayBuffer.batch_size = 32
```

## D.3  RAINBOW

### D.3.1  DEFAULT SETTINGS

Default settings used in Dopamine:

```
1  # Hyperparameters follow Hessel et al. (2018), except for sticky_actions,
2  # which was False (not using sticky actions) in the original paper.
3  import dopamine.agents.rainbow.rainbow_agent
4  import dopamine.atari.run_experiment
5  import dopamine.replay_memory.prioritized_replay_buffer
6  import gin.tf.external_configurables
7
8  RainbowAgent.num_atoms = 51
9  RainbowAgent.vmax = 10.
10 RainbowAgent.gamma = 0.99
11 RainbowAgent.update_horizon = 3
12 RainbowAgent.min_replay_history = 20000  # agent steps
13 RainbowAgent.update_period = 4
14 RainbowAgent.target_update_period = 8000  # agent steps
15 RainbowAgent.epsilon_train = 0.01
16 RainbowAgent.epsilon_eval = 0.001
17 RainbowAgent.epsilon_decay_period = 250000  # agent steps
18 RainbowAgent.replay_scheme = 'prioritized'
19 RainbowAgent.tf_device = '/gpu:0'  # use '/cpu:*' for non-GPU version
20 RainbowAgent.optimizer = @tf.train.AdamOptimizer()
21
22 # Note these parameters are different from C51's.
23 tf.train.AdamOptimizer.learning_rate = 0.0000625
24 tf.train.AdamOptimizer.epsilon = 0.00015
25
26 Runner.game_name = 'Pong'
27 # Sticky actions with probability 0.25, as suggested by (Machado et al.,
     2017).
28 Runner.sticky_actions = True
29 Runner.num_iterations = 200
30 Runner.training_steps = 250000  # agent steps
31 Runner.evaluation_steps = 125000  # agent steps
32 Runner.max_steps_per_episode = 27000  # agent steps
33
34 WrappedPrioritizedReplayBuffer.replay_capacity = 1000000
35 WrappedPrioritizedReplayBuffer.batch_size = 32
```

### D.3.2  AAAI SETTINGS

Settings used in Hessel et al. (2018):

```
1  # Hyperparameters follow Hessel et al. (2018).
2  import dopamine.atari.preprocessing
3  import dopamine.agents.rainbow.rainbow_agent
4  import dopamine.atari.run_experiment
5  import dopamine.replay_memory.prioritized_replay_buffer
6  import gin.tf.external_configurables
7
8  RainbowAgent.num_atoms = 51
9  RainbowAgent.vmax = 10.
10 RainbowAgent.gamma = 0.99
11 RainbowAgent.update_horizon = 3
12 RainbowAgent.min_replay_history = 20000  # agent steps
13 RainbowAgent.update_period = 4
14 RainbowAgent.target_update_period = 8000  # agent steps
15 RainbowAgent.epsilon_train = 0.01
16 RainbowAgent.epsilon_eval = 0.001
17 RainbowAgent.epsilon_decay_period = 250000  # agent steps
18 RainbowAgent.replay_scheme = 'prioritized'
19 RainbowAgent.tf_device = '/gpu:0'  # use '/cpu:*' for non-GPU version
20 RainbowAgent.optimizer = @tf.train.AdamOptimizer()
21
22 # Note these parameters are different from C51's.
23 tf.train.AdamOptimizer.learning_rate = 0.0000625
```

```
24 tf.train.AdamOptimizer.epsilon = 0.00015
25
26 Runner.game_name = 'Pong'
27 # Deterministic ALE version used in the AAAI paper.
28 Runner.sticky_actions = False
29 Runner.num_iterations = 200
30 Runner.training_steps = 250000  # agent steps
31 Runner.evaluation_steps = 125000  # agent steps
32 Runner.max_steps_per_episode = 27000  # agent steps
33
34 AtariPreprocessing.terminal_on_life_loss = True
35
36 WrappedPrioritizedReplayBuffer.replay_capacity = 1000000
37 WrappedPrioritizedReplayBuffer.batch_size = 32
```

### D.4   IQN

#### D.4.1   DEFAULT SETTINGS

Default settings used in Dopamine:

```
1 # Hyperparameters follow Dabney et al. (2018), but we modify as necessary
      to
2 # match those used in Rainbow (Hessel et al., 2018), to ensure apples-to-
    apples
3 # comparison.
4
5 import dopamine.agents.implicit_quantile.implicit_quantile_agent
6 import dopamine.agents.rainbow.rainbow_agent
7 import dopamine.atari.run_experiment
8 import dopamine.replay_memory.prioritized_replay_buffer
9 import gin.tf.external_configurables
10
11 ImplicitQuantileAgent.kappa = 1.0
12 ImplicitQuantileAgent.num_tau_samples = 64
13 ImplicitQuantileAgent.num_tau_prime_samples = 64
14 ImplicitQuantileAgent.num_quantile_samples = 32
15 RainbowAgent.gamma = 0.99
16 RainbowAgent.update_horizon = 3
17 RainbowAgent.min_replay_history = 20000 # agent steps
18 RainbowAgent.update_period = 4
19 RainbowAgent.target_update_period = 8000 # agent steps
20 RainbowAgent.epsilon_train = 0.01
21 RainbowAgent.epsilon_eval = 0.001
22 RainbowAgent.epsilon_decay_period = 250000  # agent steps
23 # IQN currently does not support prioritized replay.
24 RainbowAgent.replay_scheme = 'uniform'
25 RainbowAgent.tf_device = '/gpu:0'  # '/cpu:*' use for non-GPU version
26 RainbowAgent.optimizer = @tf.train.AdamOptimizer()
27
28 tf.train.AdamOptimizer.learning_rate = 0.0000625
29 tf.train.AdamOptimizer.epsilon = 0.00015
30
31 Runner.game_name = 'Pong'
32 # Sticky actions with probability 0.25, as suggested by (Machado et al.,
      2017).
33 Runner.sticky_actions = True
34 Runner.num_iterations = 200
35 Runner.training_steps = 250000
36 Runner.evaluation_steps = 125000
37 Runner.max_steps_per_episode = 27000
38
39 WrappedPrioritizedReplayBuffer.replay_capacity = 1000000
40 WrappedPrioritizedReplayBuffer.batch_size = 32
```

### D.4.2 ICML SETTINGS

Settings used in Dabney et al. (2018a):

```
1  # Hyperparameters follow Dabney et al. (2018)
2  import dopamine.agents.implicit_quantile.implicit_quantile_agent
3  import dopamine.agents.rainbow.rainbow_agent
4  import dopamine.atari.run_experiment
5  import dopamine.replay_memory.prioritized_replay_buffer
6  import gin.tf.external_configurables
7
8  ImplicitQuantileAgent.kappa = 1.0
9  ImplicitQuantileAgent.num_tau_samples = 64
10 ImplicitQuantileAgent.num_tau_prime_samples = 64
11 ImplicitQuantileAgent.num_quantile_samples = 32
12 RainbowAgent.gamma = 0.99
13 RainbowAgent.update_horizon = 1
14 RainbowAgent.min_replay_history = 50000 # agent steps
15 RainbowAgent.update_period = 4
16 RainbowAgent.target_update_period = 10000 # agent steps
17 RainbowAgent.epsilon_train = 0.01
18 RainbowAgent.epsilon_eval = 0.001
19 RainbowAgent.epsilon_decay_period = 1000000 # agent steps
20 RainbowAgent.replay_scheme = 'uniform'
21 RainbowAgent.tf_device = '/gpu:0'  # '/cpu:*' use for non-GPU version
22 RainbowAgent.optimizer = @tf.train.AdamOptimizer()
23
24 tf.train.AdamOptimizer.learning_rate = 0.00005
25 tf.train.AdamOptimizer.epsilon = 0.0003125
26
27 Runner.game_name = 'Pong'
28 Runner.sticky_actions = False
29 Runner.num_iterations = 200
30 Runner.training_steps = 250000
31 Runner.evaluation_steps = 125000
32 Runner.max_steps_per_episode = 27000
33
34 WrappedPrioritizedReplayBuffer.replay_capacity = 1000000
35 WrappedPrioritizedReplayBuffer.batch_size = 32
```

