# OpenReview forum: "Dopamine: A Research Framework for Deep Reinforcement Learning"
_ICLR.cc/2019/Conference_

### Official Review · AnonReviewer2 · 2018-10-31
**A useful framework, but may not have enough research novelty**

**Rating:** 3
**Confidence:** 3

**Review:**

Review: This paper proposed "Dopamine", a new framework for DeepRL.  While this framework seems to be useful and the paper seems like a useful guide for using the framework, I didn't think that the paper had enough scientific novelty to be an ICLR paper.  I think that papers on novel frameworks can be suitable, but they should demonstrate that they're able to do something or provide a novel capability which has not been demonstrated before.

Strengths:

-Having a standardized tool for keeping replay buffers seems useful.

-The Dopamine framework is written in Python and has 12 files, which means that it should be reasonably easy for users to understand how it's functioning and change things or debug.

-The paper has a little bit of analysis of how different settings effect results (such as how to terminate episodes) but I'm not sure that it does much to help us in understanding the framework.  I suppose it's useful to understand that the settings which are configurable in the framework affect results?

-The result on how sticky actions affect results is nice but I'm not sure what it adds over the Machado (2018) discussion.

Weaknesses:

-Given that the paper is about documenting a new framework, it would have been nice to see more comprehensive baselines documented for different methods and settings.

-I don't understand the point of 2.1, in that it seems somewhat trivial that research has been done on different architectures and algorithms.

-In section 4.2, I wonder if the impact of training mode vs. evaluation mode would be larger if the model used a stochastic regularizer.  I suspect that in general changing to evaluation mode could have a significant impact.

---

### Official Review · AnonReviewer3 · 2018-10-31
**The contribution is not ready to be published in ICLR**

**Rating:** 3
**Confidence:** 2

**Review:**

Summary:
The authors present an open-source framework TensorFlow-based named Dopamine to facilitate the task of researchers in deep reinforcement learning (deep RL). It allows to build deep RL using existing components such as reinforcement learning agents, as well as handling memory, logs and providing checkpoints for them.
Emphasis is given on providing a unified interface to these agents as well as keeping the framework generic and simple (2000 lines of code).
The framework was demonstrated on Atari games notably using Deep Q-network agents (DQN).
The authors provide numerous examples of parameter files that can be used with their framework.
Performance results are reported for some agents (DQN, C51, Rainbow, IQN).

Given the actual trends in deep learning works, unified frameworks such as that proposed is welcome.
The automatization of checkpointing for instance is particularly useful for long running experiments.
Also, trying to reduce the volume of code is beneficial for long-term maintenance and usability.

Major concerns:
* This type of contribution may not match the scope of ICLR.
* In the abstract and a large fraction of the text, the authors claim that their work is a generic reinforcement learning framework. However, the paper shows that the framework is very dependent on agents playing Atari games. Moreover, the word "Atari" comes out of nowhere on pages 2 and 5.
The authors should mention in the beginning (e.g. in the abstract) that they are handling only agents operating on Atari games.
* The positioning of the paper relative to existing approaches is unclear: state of the art is mentioned but neither discussed nor compared to the proposal.
* The format of the paper should be revised:
                - Section 5 (Related Works) should come before presenting the author's work. When reading the preceding sections, we do not know what to expect from the proposed framework.
                - All the code, especially in the appendices, seems not useful in such a paper, but rather to the online documentation of the author's framework.
* What is the motivation of the author's experiments?
                - Reproduce existing results (claimed on page 1)? Then, use the same settings as published works and show that the author's framework reaches the same level of performances.
                - Show new results (such as the effect of stickiness)? Then the authors should explicitly say that one of the contributions of the paper is to show new results.
* The authors say that they want to compare results in Figure 3. They explain why the same scale is not used. To my opinion, the authors should find a way to bring all comparisons to the same scale.

For all these reasons, I think the paper is not ready for publication at ICLR.

---

### Official Review · AnonReviewer1 · 2018-11-03
**Needs refinement**

**Rating:** 3
**Confidence:** 4

**Review:**

This paper introduces and details a new research framework for reinforcement learning called Dopamine. The authors give a brief description of the framework, built upon Tensorflow, and reproduce some recent results on the ALE framework.

Pros:
1. Nice execution and they managed to successfully reproduce recent deep RL results, which can be challenging at times.

Cons:
1. Given that this is a paper describing a new framework, I expected a lot more in terms of comparing it to existing frameworks like OpenAI Gym, RLLab, RLLib, etc. along different dimensions.  In short, why should I use this framework? Unfortunately, the current version of the paper does not provide me information to make this choice. Other than the framework, the paper does not present any new tasks/results/algorithms, so it is not clear what the contribution is.


Other comments:
1. The paragraphs in sections 2.1 and 2.2 (algorithmic research, architecture research, etc.) seem to say pretty much the same things. They could be combined, and the DQN can be used as a running example to make the points clear.
2. The authors mention tests to ensure reliability and reproducibility. Can you provide more details? Do these tests account for semantic bugs common while implementing RL algorithms?

---

### Author Response · Authors · 2018-11-06
**Response to reviews**

We would like to thank all the reviewers for their comments.

We feel ICLR is the right venue for this type of contribution, as it is providing a stable, reproducible, and reliable framework for others to use.
Similar frameworks have been previously introduced at comparable conferences: ELF at NIPS 2017 and RLLib at ICML 2018.

---

> ### Comment · AnonReviewer2 · 2018-11-07
> **Thanks**
>
> My feeling towards this is that I think it's perfectly reasonable for ICLR to publish new frameworks.  But my view is that the contribution needs to entail a novel capability (i.e. it lets us do something that we couldn't do before, or that would be very hard to do before) as opposed to a well-executed framework that does things that have already been doable.
>
> For example, there are strengths to having a framework which is self-contained, but does it provide new capabilities?
>
> This is just my perspective, apparently the ELF paper got a similar review, but the reviewer changed their mind after comment from the area chair / rebuttal (which we can't see):
>
> https://media.nips.cc/nipsbooks/nipspapers/paper_files/nips30/reviews/1522.html

---

### Meta-Review · Area_Chair1 · 2018-12-15
**Supportive of open source DRL frameworks, but this is not a scientific contribution**

**Confidence:** 3
**Recommendation:** Reject

**Metareview:**

The paper presents Dopamine, an open-source implementation of plenty of DRL methods. It presents a case study of DQN and experiments on Atari. The paper is clear and easy to follow.

While I believe Dopamine is a very welcomed contribution to the DRL software landscape, it seems there is not enough scientific content in this paper to warrant publication at ICLR. Regarding specifically the ELF and RLlib papers, I think that the ELF paper had a novelty component, and presented RL baselines to a new environment (miniRTS), while the RLlib paper had a stronger "systems research" contribution. This says nothing about the future impact of Dopamine, ELF, and RLlib – the respective software.